# Building Trust and Enhancing Tax Compliance: The Role of Authoritarian Procedures and Respectful Treatment in Indonesia

**Dewi Prastiwi [1,*] and Erlina Diamastuti [2]**

1  Accounting Department, Faculty of Economic, Universitas Negeri Surabaya, Surabaya 60231, Indonesia
2  Accounting Department, Universitas Internasional Semen Indonesia, Gresik 61122, Indonesia; erlina.diamastuti@uisi.ac.id
*  Correspondence: dewiprastiwi@unesa.ac.id; Tel.: +62-31-99421835

**Abstract:** This study delves into the impact of tax collection behavior on tax compliance among individual taxpayers in Indonesia, with a specific focus on two distinct behaviors: respectful treatment and authoritarian procedures. The research employs a cross-sectional survey method, targeting the population of individual taxpayers registered at the Regional Tax Office of East Java I. The sample size of 400 was selected through random sampling. Attitudes, opinions, and perceptions regarding tax collection behavior were measured using a Likert scale. Tax officials' conduct was categorized as either respectful treatment or authoritarian procedures. The research employed Structural Equation Modeling (SEM) with the Partial Least Squares (PLS) software to assess the outer model. Hypothesis testing was conducted to scrutinize the relationship between tax collection behavior and taxpayer compliance. The study's results indicate that respectful treatment positively influences compliance, whereas the utilization of authoritarian procedures leads to an increase in tax non-compliance. Notably, trust emerged as a mediating factor within this relationship. The findings underscore the crucial role of tax officials in cultivating trust with taxpayers by demonstrating respect, upholding integrity, and executing their responsibilities transparently and equitably. By fostering an environment of trust, tax compliance can be bolstered, fostering a collaborative approach that aids taxpayers in fulfilling their tax obligations.

**Keywords:** tax collecting behavior; tax compliance; respectful treatment; authoritarian procedures; trust mediation

## 1. Introduction

The concept of behavior has become integral to public policy on a broader scale (Gopalan and Pirog 2017). The behavioral dimension of authorities holds significant importance in shaping and implementing policies. It has the potential to establish an environment that is more inclusive, transparent, and accountable. This, in turn, can contribute to heightened policy legitimacy, increased public participation, improved policy effectiveness, and the prevention of corruption. This principle is equally applicable to tax policy. Past research underscores a sociological approach that places emphasis on the behavioral facets of tax officials within the context of tax system cooperation proves to be an effective strategy for enhancing taxpayer compliance (Alm et al. 2010; Battiston and Gamba 2016; Castro and Scartascini 2015; Eichfelder and Kegels 2014; Gangl et al. 2014; Mohdali et al. 2014; Vossler and McKee 2017).

Earlier studies indicate that under certain circumstances, external interventions like monetary incentives or punishments could have an adverse effect on tax compliance (Battiston and Gamba 2016; Gangl et al. 2014; De Neve et al. 2020; Schächtele et al. 2023). The interplay between taxpayers and tax officials represents a longstanding psychological contract, where taxpayer compliance is influenced by the mutual cooperation of these

stakeholders within the tax system (Prastiwi et al. 2021). A tax system that nurtures a harmonious atmosphere proves to be a potent tool in enhancing tax compliance. The treatment of taxpayers solely as subjects to enforce tax payment often leads to tax avoidance. The psychological contract between taxpayers and tax authorities exerts an impact on the level of tax morale, thereby addressing non-compliance concerns (Aulia et al. 2022; Gangl et al. 2019).

A significant issue in Indonesia pertains to the conduct of tax officials, particularly in terms of abuse of power. Some tax officials might misuse their authority when enforcing tax laws, resorting to tactics like intimidation, extortion, or undue pressure on taxpayers. Such misuse of power not only harms taxpayers but also perpetuates injustice within the tax system, subsequently affecting tax compliance. An authoritarian approach from tax officials could erode tax morale, especially when taxpayers are already compliant. Conversely, a courteous, amicable approach or a responsive regulatory strategy can nurture tax morale, particularly when taxpayers already exhibit a high level of compliance (Khan and Nuryanah 2023; Rahman 2017). Employing threats and coercion tends to yield counter-productive behavior (Goltz 2020). By treating taxpayers with respect, rendering services, and adopting a humane approach (respectful treatment behavior) when communicating their tax obligations and addressing tax avoidance, tax authorities can foster trust among taxpayers, leading to accurate reporting of tax dues and increased tax compliance (Farrar et al. 2019; Gangl and Torgler 2020). An excessive focus on preventive measures could potentially breed taxpayer skepticism, whereas respectful treatment has a positive impact on boosting tax morale (Weber et al. 2014).

This study undertakes an examination of how the behavior of tax collectors influences taxpayer compliance in Indonesia. It considers the cultural context of hierarchy and respect for elders that prevails in the country. The investigation concentrates on two distinct behaviors: respectful treatment and authoritarian procedures. Additionally, the research explores whether trust plays a mediating role in how respectful treatment impacts compliance. The primary objective is to furnish evidence supporting the effectiveness of treating taxpayers respectfully as a means to enhance compliance, thereby moving away from a reliance solely on punitive measures. Ultimately, the study endeavors to contribute to the enhancement of tax policies in Indonesia by incorporating the behavior of tax authorities in fostering collaboration with taxpayers.

## 2. Hypotheses

Compliance plays a pivotal role in the success of tax collection efforts. Multiple studies have examined factors influencing tax compliance from diverse perspectives and theoretical frameworks. One common theoretical framework often employed by researchers is the economic crime model proposed by Becker (Becker 1968). According to this approach, taxpayers are viewed as rational actors who weigh the costs and benefits associated with their behaviors (Feld and Larsen 2012). Taxpayers are faced with a trade-off between the potential financial gains of tax avoidance and the potential repercussions of detection and tax evasion, including penalties (Castro and Scartascini 2015). In this perspective, tax compliance is shaped by the perceived risk of being caught by tax authorities and the severity of the punishment for violating tax regulations (Alm et al. 2010; Alm et al. 2009; Cullis and Lewis 1997; Feld and Larsen 2012; Gangl et al. 2014; Kleven et al. 2011; Lisi 2015; Luttmer and Singhal 2014; McKee et al. 2018; Mohdali et al. 2014; Slemrod and Yitzhaki 2002; Vossler and McKee 2017).

However, an alternative approach to enhancing taxpayer compliance, known as the sociological approach, has also gained traction. This approach posits that tax compliance is influenced by the cooperation between the parties within the tax system. The relationship between taxpayers and tax authorities is a psychological one, built on mutual interdependence. Tax authorities aim to maximize net revenue while minimizing tax collection costs, which encompass not only audit expenses but also education and communication costs. As

such, tax authorities can choose the most efficient methods of dealing with taxpayers to reduce collection costs (Langham et al. 2012).

Indeed, respectful treatment from tax authorities can positively influence tax morale. Conversely, when taxpayers are treated merely as subjects compelled to pay taxes, they are more inclined to resort to tax avoidance behaviors (Frey and Feld 2002). This relationship can be seen as an implicit or relational contract between taxpayers and tax authorities, characterized by emotional bonds and loyalty. Studies have shown that incentives and respectful treatment are key factors affecting tax compliance. Greater respect shown by tax authorities to deter tax avoidance efforts translates into higher taxpayer willingness to fulfill their tax obligations, leading to reduced levels of tax avoidance.

Efforts aimed at respectful prevention can take two forms. Firstly, audit procedures employed by tax authorities must be transparent and clear. Arbitrary actions weaken taxpayers' positions and erode their motivation to pay taxes. Secondly, respectful treatment has a direct personal impact on how taxpayers perceive their treatment by tax officials. Treating taxpayers as partners in the psychological tax contract, rather than as subordinate entities in hierarchical relationships, encourages honest tax payment. Additionally, respectful treatment reinforces emotional effects on compliance behavior.

In accordance with the framework presented by Feld and Frey (2002b), two contrasting methods of treating taxpayers emerge, the authoritarian procedure and respectful treatment. The authoritarian procedure entails tax authorities employing threats to remind taxpayers of their tax obligations and discourage tax avoidance. Errors in tax reporting are met with immediate suspicion of fraud and potential legal sanctions. Conversely, respectful treatment involves tax authorities engaging taxpayers in transparent, non-threatening dialogue. This respectful approach positively impacts tax morale. Characteristics of these approaches include how procedures are determined (from the perspective of authorities for authoritarian procedure, and in a transparent, respectful manner for respectful treatment), and how taxpayers are treated (dictated by authorities for authoritarian procedure, and as partners in a psychological contract for respectful treatment).

Trust plays a pivotal role in shaping this relationship. Trust in tax authorities can enhance their authoritarian power. For example, increased trust can encourage individuals to report tax evasion cases. Conversely, a decrease in trust can erode this authority. Power dynamics and trust are intricately linked within the Slippery Slope Framework (Kirchler et al. 2008).

The Directorate General of Taxes' exercise of legitimacy power through the approach of respectful treatment can demonstrate its ability to resolve taxpayers' issues with goodwill and integrity, building trust and ultimately bolstering compliance. Therefore, this study hypothesizes that trust acts as a mediator between respectful treatment and taxpayer compliance.

**H1.** *Respectful treatment has a positive effect on taxpayer compliance.*

**H2.** *Authoritarian procedure has a positive effect on taxpayer non-compliance.*

**H3.** *Trust mediates the relationship between respectful treatment and taxpayers' compliance.*

## 3. Materials and Methods

In this study, the research methodology employed a cross-sectional survey approach. The research population comprised individual taxpayers registered at the Regional Tax Office of East Java I. Individual taxpayers were selected as the research subjects due to their direct representation of taxpayers' compliance attitudes, as opposed to corporate taxpayers. The study sample was a subset of this population, with the sampling technique employed being random sampling. Random sampling involves the random selection of respondents without specific criteria, ensuring that all taxpayers within the jurisdiction of the East Java I Regional Tax Office have an equal chance of being included. The total population of individual taxpayers under the East Java I Regional Tax Office in 2022 was 1,143,203.

The determination of the sample size in this study followed Slovin's Formula, considering a confidence level of 5%. This calculation yielded a sample size of 400 individuals. The sampling of taxpayers within the East Java I Regional Tax Office was conducted using a relative calculation method. This method involved comparing the percentage of taxpayers in each Tax Service Office (KPP) to the total number of taxpayers under the jurisdiction of the East Java I Regional Tax Office.

The data collection process involved distributing questionnaires directly to the respondents during the Annual Tax Return filing period, which occurred between February and March 2023. These questionnaires were distributed at the Tax Service Offices falling within the purview of the East Java I Regional Tax Office. The intended number of respondents from the East Java I Regional Tax Office is outlined in Table 1 below.

**Table 1.** The number of respondents.

| KPP DJP East Java I | Number of Taxpayers | Relative Percentage | Target Taxpayers |
|---|---|---|---|
| KPP Pratama Surabaya Mulyorejo | 241,560 | 21.13% | 85 |
| KPP Pratama Surabaya Sukomanunggal | 158,948 | 13.90% | 56 |
| KPP Pratama Surabaya Wonocolo | 153,370 | 13.42% | 54 |
| KPP Pratama Surabaya Sawahan | 123,863 | 10.83% | 43 |
| KPP Pratama Surabaya Gubeng | 120,510 | 10.54% | 42 |
| KPP Pratama Surabaya Rungkut | 106,968 | 9.36% | 37 |
| KPP Pratama Surabaya Karangpilang | 90,158 | 7.89% | 32 |
| KPP Pratama Surabaya Tegalsari | 42,408 | 3.71% | 15 |
| KPP Pratama Surabaya Krembangan | 41,740 | 3.65% | 15 |
| KPP Pratama Surabaya Genteng | 32,249 | 2.82% | 11 |
| KPP Pratama Surabaya Pabean Cantikan | 31,429 | 2.75% | 11 |
| Total | 1,143,203 | 100.00% | 400 |

This study examines how tax collection behavior influences tax compliance among individual taxpayers in Indonesia, differentiating between two distinct categories: respectful treatment and authoritarian procedure. These behaviors' associated attitudes, viewpoints, and perceptions are evaluated using a Likert scale, ranging from 1 (strongly disagree) to 10 (strongly agree). Tax officials are classified as exhibiting respectful treatment if their actions involve clarifying errors through personal communication, assisting taxpayers in overcoming challenges, and prioritizing ease within the tax process. Conversely, tax officials are identified as employing authoritarian procedures when their actions contrast with the above. The dependent variable, tax compliance, is measured through indicators like voluntary tax payment, complete income disclosure, and prompt reporting of tax obligations. Furthermore, the study delves into the role of trust as a mediating variable, gauged through indicators including belief in the tax authority's competence to manage taxes, the perception of their benevolent intentions for the common good, and trust in the integrity of tax officials (Aktaş Güzel et al. 2019; Ali and Ahmad 2014; Bornman 2015; Kirchler et al. 2008; Muehlbacher et al. 2011).

The research methodology employs the Structural Equation Modeling (SEM) approach, utilizing the Partial Least Squares (PLS) software to test the outer model. The assessment of the outer model entails evaluating its validity and reliability. Validity testing determines the capacity of the measurement instrument to accurately gauge its intended constructs. Conversely, reliability testing examines the consistency of the measurement tool or respondents' responses in measuring a concept. The outer model's assessment encompasses convergent validity, discriminant validity, and composite reliability tests.

For hypothesis testing within the inner model, respondent answers are categorized based on their perception of tax collection behavior, namely authoritarian procedure and respectful treatment. This categorization hinges on the average scores respondents give to the question indicators. Respondents fall into the authoritarian procedure category if their average scores for the indicators are < 8 or at most 7.9. On the other hand, they are categorized as experiencing respectful treatment if their average scores for the indicators exceed 8. The relationship between independent and dependent variables is subsequently tested using SPSS version 25.

Throughout this research project, a strong commitment to ethical research conduct is upheld, safeguarding the rights and privacy of participants. All research activities adhere to the guidelines and regulations established by Universitas Negeri Surabaya's Ethics Committee. Participants were provided with transparent information about the research's purpose, procedures, and potential risks and benefits. Participation was voluntary, and participants retained the prerogative to withdraw at any point without incurring consequences. Confidentiality of personal information was maintained, and collected data were solely used for research purposes.

## 4. Results

### 4.1. Descriptive Statistics and Model Evaluation (Outer and Inner)

Table 2 displays the distribution of the sample across different demographic characteristics: age (Panel A), gender (Panel B), education (Panel C), occupation (Panel D), and annual income level (Panel E).

In Panel A, the respondents' ages are categorized into six groups, each spanning a 10-year range. The largest proportion of respondents falls within the age group of 21–30 (158), constituting 39.5% of the sample. Following this, the age group of 31–40 encompasses 83 respondents, accounting for 20.7%. The age group of 41–50 includes 96 respondents, representing 24%. The remaining respondents are either below 20 years old or over 50 years old.

Panel B showcases the gender distribution among the respondents. The majority of participants in the study are female (215), making up 53.7% of the sample, while the remainder are male.

Turning to Panel C, respondents' education levels are categorized as having a bachelor's degree or non-bachelor's degree education. Those with a bachelor's degree account for a dominant share (220), constituting 55%, whereas the number of respondents with non-bachelor's degree education stands at 180.

In Panel D, respondents' occupations are divided into two groups. The largest group comprises employees (283), while the second group consists of other occupations.

Panel E illustrates the distribution of respondents' annual income levels. A total of 289 respondents have an annual income between IDR 0 and 60,000,000. There are 92 respondents falling within the income range of IDR 60,000,000 to 250,000,000 per year. Additionally, 17 respondents report an annual income between IDR 250,000,000 and 500,000,000, while only 2 respondents have an annual income ranging from IDR 500,000,000 to 5,000,000,000. This categorization aligns with the income tax rate structure.

These descriptive statistics provide a comprehensive snapshot of the sample's composition based on age, gender, education, occupation, and income level. The data offer valuable insights into the demographic profile of the study's participants within each category.

**Table 2.** Sample distribution.

| Panel A: Sample Distribution Based on Age | | | | | | |
|---|---|---|---|---|---|---|
| **Age (years)** | **Authoritarian Procedure** | | **Respectful Treatment** | | **Total** | |
| | **N** | **%** | **N** | **%** | **N** | **%** |
| < 20 | 0 | 0.00 | 3 | 1.43 | 3 | 0.75 |
| 21–30 | 75 | 39.47 | 83 | 39.52 | 158 | 39.50 |
| 31–40 | 44 | 23.16 | 39 | 18.57 | 83 | 20.75 |
| 41–50 | 50 | 26.32 | 46 | 21.90 | 96 | 24.00 |
| 51–60 | 15 | 7.89 | 35 | 16.67 | 50 | 12.50 |
| 61–70 | 6 | 3.16 | 4 | 1.90 | 10 | 2.50 |
| Total | 190 | 100 | 210 | 100 | 400 | 100 |

| Panel B: Sample Distribution Based on Gender | | | | | | |
|---|---|---|---|---|---|---|
| **Gender** | **Authoritarian Procedure** | | **Respectful Treatment** | | **Total** | |
| | **N** | **%** | **N** | **%** | **N** | **%** |
| Male | 82 | 43.16 | 103 | 49.05 | 185 | 46.25 |
| Female | 108 | 56.84 | 107 | 50.95 | 215 | 53.75 |
| Total | 190 | 100 | 210 | 100 | 400 | 100 |

| Panel C: Sample Distribution Based on Education | | | | | | |
|---|---|---|---|---|---|---|
| **Education** | **Authoritarian Procedure** | | **Respectful Treatment** | | **Total** | |
| | **N** | **%** | **N** | **%** | **N** | **%** |
| Bachelor's Degree | 107 | 56.32 | 113 | 53.81 | 220 | 55 |
| Non-Bachelor's Degree | 83 | 43.68 | 97 | 46.19 | 180 | 45 |
| Total | 190 | 100 | 210 | 100 | 400 | 100 |

| Panel D: Sample Distribution Based on Occupation | | | | | | |
|---|---|---|---|---|---|---|
| **Occupation** | **Authoritarian Procedure** | | **Respectful Treatment** | | **Total** | |
| | **N** | **%** | **N** | **%** | **N** | **%** |
| Employee | 140 | 73.68 | 68.10 | 68.1 | 283 | 70.75 |
| Non-Employee | 50 | 26.32 | 31.90 | 31.9 | 117 | 29.25 |
| Total | 190 | 100 | 210 | 100 | 400 | 100 |

| Panel E: Sample Distribution Based on Income Level (in IDR = Indonesian Rupiah) | | | | | | |
|---|---|---|---|---|---|---|
| **Income (IDR)** | **Authoritarian Procedure** | | **Respectful Treatment** | | **Total** | |
| | **N** | **%** | **N** | **%** | **N** | **%** |
| 0–60,000,000 | 134 | 70.53 | 155 | 73.81 | 289 | 72.25 |
| 60,000,000–250,000,000 | 45 | 23.68 | 47 | 22.38 | 92 | 23.00 |
| 250,000,000–500,000,000 | 9 | 4.74 | 8 | 3.81 | 17 | 4.25 |
| 500,000,000–5,000,000,000 | 2 | 1.05 | 0 | 0.00 | 2 | 0.50 |
| Total | 190 | 100 | 210 | 100 | 400 | 100 |

| Panel F: Sample Distribution Based on Registered Region | | | | | | |
|---|---|---|---|---|---|---|
| **Registered as Taxpayer at the Tax Office** | **Authoritarian Procedure** | | **Respectful Treatment** | | **Total** | |
| | **N** | **%** | **N** | **%** | **N** | **%** |
| KPP Pratama Genteng | 1 | 0.53 | 1 | 0.48 | 2 | 0.50 |
| KPP Pratama Gubeng | 22 | 11.58 | 28 | 13.33 | 50 | 12.50 |
| KPP Pratama Karangpilang | 23 | 12.11 | 34 | 16.19 | 57 | 14.25 |
| KPP Pratama Krembangan | 9 | 4.74 | 14 | 6.67 | 23 | 5.75 |
| KPP Pratama Mulyorejo | 39 | 20.53 | 43 | 20.48 | 82 | 20.50 |
| KPP Pratama Rungkut | 23 | 12.11 | 22 | 10.48 | 45 | 11.25 |
| KPP Pratama Sawahan | 11 | 5.79 | 16 | 7.62 | 27 | 6.75 |
| KPP Pratama Sukomanunggal | 40 | 21.05 | 33 | 15.71 | 73 | 18.25 |
| KPP Pratama Tegalsari | 9 | 4.74 | 11 | 5.24 | 20 | 5.00 |
| KPP Pratama Wonocolo | 13 | 6.84 | 8 | 3.81 | 21 | 5.25 |
| Total | 190 | 100 | 210 | 100 | 400 | 100 |

### 4.2. Evaluation of Outer Model

Table 3 presents the outcomes of the validity and reliability assessments. In Panel A, validity testing includes both convergent validity and discriminant validity evaluations. Convergent validity is determined based on substantial and satisfactory outer loadings, which are expected to surpass 0.7 (Hair et al. 2011; Latan and Ramli 2013; Mardiana and Faqih 2019). Discerning discriminant validity relies on the fornell–larcker criterion and the cross loading table, which demand that the latent construct's value should surpass the corresponding values of other variables (Hair et al. 2011; Latan and Ramli 2013; Sarstedt et al. 2014). The data in Panel A of Table 3 indicate that all indicators of the variables meet the validity criteria, confirming the validity of the study's variables.

**Table 3.** Evaluation results of outer model.

**Panel A: Validity Scores (Convergent Validity and Discriminant Validity)**

| Indicator of Variable | Convergent Validity | | | Discriminant Validity | | | | | | |
|---|---|---|---|---|---|---|---|---|---|---|
| | Outer Loadings | | | Cross Loadings | | | Fornell–Larcker Criterion | | | |
| | TC | TCB | TR | TC | TCB | TR | | TC | TCB | TR |
| X1-2 | - | 0.702 | - | 0.348 | 0.702 | 0.433 | TC | 0.886 | - | - |
| X1-3 | - | 0.737 | - | 0.333 | 0.737 | 0.445 | TCB | 0.500 | 0.752 | - |
| X1-5 | - | 0.729 | - | 0.372 | 0.729 | 0.446 | TR | 0.490 | 0.649 | 0.845 |
| X1-6 | - | 0.753 | - | 0.336 | 0.753 | 0.440 | | | | |
| X1-7 | - | 0.769 | - | 0.318 | 0.769 | 0.445 | | | | |
| X1-8 | - | 0.753 | - | 0.407 | 0.753 | 0.469 | | | | |
| X1-9 | - | 0.787 | - | 0.391 | 0.787 | 0.581 | | | | |
| X1-10 | - | 0.782 | - | 0.469 | 0.782 | 0.594 | | | | |
| TR1 | - | - | 0.756 | 0.542 | 0.579 | 0.756 | | | | |
| TR2 | - | - | 0.869 | 0.339 | 0.509 | 0.869 | | | | |
| TR3 | - | - | 0.906 | 0.351 | 0.528 | 0.906 | | | | |
| TR4 | - | - | 0.856 | 0.410 | 0.552 | 0.856 | | | | |
| TR5 | - | - | 0.833 | 0.383 | 0.545 | 0.833 | | | | |
| Y-1 | 0.843 | - | - | 0.843 | 0.446 | 0.468 | | | | |
| Y-2 | 0.884 | - | - | 0.884 | 0.448 | 0.426 | | | | |
| Y-3 | 0.920 | - | - | 0.920 | 0.424 | 0.410 | | | | |
| Y-4 | 0.908 | - | - | 0.908 | 0.421 | 0.412 | | | | |
| Y-5 | 0.874 | - | - | 0.874 | 0.469 | 0.448 | | | | |

**Panel B: Reliability value (Composite Reliability)**

| | Cronbach's Alpha | rho_A | Composite Reliability | Average Variance Extracted (AVE) |
|---|---|---|---|---|
| TCB | 0.891 | 0.896 | 0.912 | 0.566 |
| TR | 0.899 | 0.901 | 0.926 | 0.715 |
| TC | 0.931 | 0.932 | 0.948 | 0.785 |

In Panel B, the results of the reliability assessment establish the reliability of the variables under scrutiny, as they adhere to the reliability test criteria. The reliability of a construct can be gauged using two approaches: Cronbach's alpha and composite reliability. However, due to the tendency of Cronbach's alpha to yield lower values, it is advisable to utilize composite reliability. The benchmark for an acceptable composite reliability score is greater than 0.7, corroborated by a Cronbach's alpha value surpassing 0.7 (Hair et al. 2011, 2014; Latan and Ramli 2013; Sarstedt et al. 2014).

### 4.3. Inner Model Evaluation

Table 4 provides an overview of the descriptive statistics pertaining to the variables: authoritarian procedure (Panel A) and respectful treatment (Panel B). In Panel A, the mean value for the authoritarian procedure variable is recorded at 69.5%, signifying the average response level from participants for this construct. Meanwhile, the mean value for tax compliance is noted as 83.5%. Shifting to Panel B, the mean value for the respectful

treatment variable stands at 87.2%. This represents the central tendency of participant responses towards respectful treatment. Furthermore, the mean value for tax compliance is established as 92%, while the mean value for the mediating variable trust registers at 85.2%.

**Table 4.** Descriptive statistics.

| Panel A: Authoritarian Procedure Testing | | | |
|---|---|---|---|
| Variable | Mean | Minimum | Maximum |
| AP | 6.9479 | 1.50 | 7.90 |
| TC | 8.3495 | 2.80 | 10.00 |
| **Panel B: Respectful Treatment Testing** | | | |
| Variable | Mean | Minimum | Maximum |
| RT | 8.7191 | 8.00 | 10.00 |
| TR | 8.5238 | 3.00 | 10.00 |
| TC | 9.1957 | 6.40 | 10.00 |

Table 5 presents the test outcomes for various relationships and effects in the study. In Panel A, the results of testing the variable "respectful treatment towards tax compliance" on tax compliance are presented. The R-squared value of 0.342 signifies that 34.2% of the variability in tax compliance can be predicted using the "respectful treatment" variable. The significant impact of "respectful treatment" on tax compliance is evident, with a significance value below 0.05. Proceeding to Panel B, the testing results indicate that the variable "authoritarian procedure" towards taxpayer non-compliance has an R-squared value of 0.297. This indicates that 29.7% of the variance in taxpayer non-compliance can be accounted for by the "authoritarian procedure" variable. Furthermore, the variable "authoritarian procedure" significantly influences taxpayer non-compliance, as demonstrated by its significance value being below 0.05. In Panel C, the testing results focus on the variables "respectful treatment" and trust in relation to taxpayer compliance. The R-squared value for the combined influence of the "respectful treatment" variable and the trust variable on taxpayer compliance is calculated as 0.406, indicating that these variables together can predict 40.6% of the variability in taxpayer compliance. Both the "respectful treatment" variable and the trust variable significantly influence taxpayer compliance, with significance values below 0.05.

The mediating effect is then calculated using the Sobel test. The calculation involves several steps and computations of standard errors. The t-statistic values for the mediation effects are determined based on these calculations. In this specific case, the t-statistic value for the mediation effect is found to be 65.7304. The results of the Sobel test calculation reveal that the variable "trust" partially mediates the relationship between the "respectful treatment" variable and the "tax compliance" variable. This indicates that trust plays a role in connecting respectful treatment with tax compliance but does not solely account for the relationship.

The mediating effect is calculated using the Sobel test as follows:

$$Sp2p3 = \sqrt{p3^2 Sp2^2 + p2^2 Sp3^2 + Sp2^2 Sp3^2}$$

$$Sp2p3 = \sqrt{(0.172)^2(0.113)^2 + (0.855)^2(0.05)^2 + (0.113)^2(0.05)^2}$$

$$Sp2p3 = \sqrt{(0.000378) + (0.0018) + (0.000032)}$$

$$Sp2p3 = 0.002237$$

The t-statistic values for the mediation effects are as follows:

$$t = \frac{p2p3}{Sp2Sp3} = \frac{0.14706}{0.002237} = 65.7304$$

**Table 5.** Test results.

| Panel A: Respectful Treatment towards tax compliance | | | | | |
|---|---|---|---|---|---|
| | R Square | | | | |
| RT | R | R Square | Adjusted R Square | Std. Error of the Estimate | |
| | 0.342 | 0.117 | 0.113 | 0.70704 | |
| | *t*-test | | | | |
| | Unstandardized Coefficients | | Standardized Coefficients Beta | | |
| RT | B | Std. Error | | t | Sig. |
| | 0.439 | 0.084 | 0.342 | 5.252 | 0.000 |
| **Panel B: Authoritarian Procedure on taxpayer non-compliance** | | | | | |
| | R Square | | | | |
| AP | R | R Square | Adjusted R Square | Std. Error of the Estimate | |
| | 0.297 | 0.088 | 0.083 | 1.18924 | |
| | *t*-test | | | | |
| | Unstandardized Coefficients | | Standardized Coefficients Beta | | |
| AP | B | Std. Error | | t | Sig. |
| | 0.388 | 0.091 | 0.297 | 4.263 | 0.000 |
| **Panel C: Respectful Treatment on tax compliance through trust** | | | | | |
| | R Square | | | | |
| RT, TR | R | R Square | Adjusted R Square | Std. Error of the Estimate | |
| | 0.406 | 0.165 | 0.157 | 0.68916 | |
| | *t*-test | | | | |
| | Unstandardized Coefficients | | Standardized Coefficients Beta | | |
| RT | B | Std. Error | | t | Sig. |
| | 0.388 | 0.091 | 0.297 | 4.263 | 0.000 |
| TR | 0.172 | 0.050 | 0.247 | 3.454 | 0.001 |

## 5. Discussion

### 5.1. Respectful Treatment and Tax Compliance

Table 5, Panel A, confirms the acceptance of Hypothesis 1. The results indicate that "respectful treatment" within tax collection behavior can significantly enhance tax compliance by 5%. This outcome is in line with findings from prior studies (Abbiati et al. 2020; Battiston and Gamba 2016; Eichfelder and Kegels 2014; De Neve et al. 2020) that discovered tax officials' behavior significantly influences tax compliance. Tax compliance is not solely determined by tax laws but also by their enactment through tax authorities. Complex regulations, legal language variations, and other uncontrollable constraints can hinder taxpayers in fulfilling their obligations. Frequent changes in tax regulations and interpretations of rules can lead to varying reporting decisions, sometimes mistaken for tax avoidance or evasion due to misunderstandings. Tax officials who adopt a helpful, cooperative approach can build positive relationships with taxpayers. Their assistance, guidance, and efficiency can alleviate taxpayer fears, increasing compliance (Gangl et al. 2015; De Neve et al. 2020).

The tax relationship establishes a long-term contract intertwined with psychological dynamics. Cultivating a positive rapport requires an active role for the tax authority, extending beyond enforcement to public service and community integration (Hlastec et al. 2023; Maldonado Valera et al. 2022). Embracing transparency, respect, and support fosters voluntary compliance by diminishing the social distance between the tax authority and

the taxpayers (Gangl et al. 2015; Slemrod 2019). Nudge theory integration augments this framework by subtly influencing behavior through public policy, underscoring the broader implications of tax policies and economic growth (Espinosa et al. 2022). When tax authorities treat taxpayers with respect, such as clarifying errors personally, prioritizing assistance, and avoiding threats, they enhance tax morale. This relational approach fosters loyalty and emotional bonds, promoting compliance (Abbiati et al. 2020; Gangl et al. 2015).

Incentives and rewards in the tax system significantly impact innovative entrepreneurship. The tax-neutral principle, as highlighted with the Laffer curve, underscores the relationship between tax rates and revenue generation, emphasizing the potential negative impact of excessive taxation (Laffer 2004). Dan Mitchell's research on economic growth through low tax rates reinforces the notion that excessive taxation is detrimental, especially for developing economies (Edwards and Mitchell 2008). By incorporating these concepts, a comprehensive understanding of tax policies aligning with economic growth emerges.

### 5.2. Authoritarian Procedure and Tax Non-Compliance

Table 5, Panel B, confirms the acceptance of hypothesis 2. The results indicate that "authoritarian procedure" within tax collection behavior can significantly increase tax non-compliance by 5%. This aligns with the findings of (Feld and Frey 2002a), who stated that when taxpayers are treated as subjects who are forced to pay taxes, it leads to tax avoidance. Similar viewpoints have been expressed by some previous studies (Battiston and Gamba 2016; Frey and Jegen 2001; Feld and Larsen 2012; Gangl et al. 2014; Goltz 2020), suggesting that external interventions, especially punitive measures, can negatively impact tax compliance under specific conditions. The authoritarian approach is considered too limited in comprehending taxpayer compliance (Dulleck et al. 2016).

The "authoritarian procedure" involves reminding taxpayers of their obligations and reducing tax avoidance through threats. If errors in tax reporting occur, tax authorities tend to immediately suspect fraudulent intent and apply legal sanctions. Alternatively, tax officials can initially inquire about the reasons behind such errors. However, if taxpayers perceive that they are suspected of tax evasion and are treated disrespectfully, their intrinsic motivation to comply decreases, and tax morale wanes. Authoritarian treatment by tax officials can generate fear and intimidation among taxpayers (Lewin et al. 1939; Prastiwi et al. 2021; Sallai and Schnyder 2020). This may lead taxpayers to avoid interactions or even attempt to evade taxes entirely.

Such authoritarian behavior can trigger distrust in the tax system, causing taxpayers to question its integrity and fairness. This can diminish their motivation to comply, as well as their willingness to provide necessary tax information. Fearful or distrustful taxpayers may withhold information, obstructing tax enforcement and reducing revenue (Chen and Zhang 2021; Dodlova and Lucas 2021; Gilley 2017). This research provides insights into how respectful treatment and authoritarian procedure within tax collection behavior significantly impact tax compliance and non-compliance, respectively. Respectful treatment fosters positive relationships, trust, and compliance, while authoritarian procedures can trigger avoidance behaviors and reduce tax morale. These findings contribute to the ongoing understanding of tax compliance factors and have implications for designing effective tax policies and enforcement strategies.

### 5.3. Trust Can Mediate the Relationship between Respectful Treatment and Tax Compliance

Table 5, Panel C, validates hypothesis 3, indicating that the relationship between "respectful treatment" and tax compliance is partially mediated by trust. This result aligns with the Slippery Slope Framework theory, which integrates social psychology tax models and posits that tax compliance can be enhanced through increased power and trust (Gangl et al. 2014; Gangl and Torgler 2020; Kirchler et al. 2008). The authoritarian procedure can be interpreted as a signal of distrust in taxpayers' honesty, while a fair approach fosters trust and compliance (Kirchler et al. 2008; Vossler and McKee 2017; Weber et al. 2014).



The study's findings are in line with previous studies (Alm and Torgler 2011; Kirchler et al. 2008), suggesting that respectful treatment by tax authorities, along with services and formal procedures, enhances taxpayers' trust, which leads to the accurate reporting of tax obligations and subsequently improves tax compliance. The provision of good service and support in fulfilling tax obligations can strengthen taxpayers' feelings of being valued, leading to greater motivation to comply with regulations (Asamoah 2018).

The way taxes are managed and the behavior of tax collectors can have a significant impact on corruption and, consequently, economic growth. If tax policies are poorly designed and tax collectors exhibit negative attitudes, corruption can take root, leading to harmful effects on the economy. Corruption diverts resources, hampers investments, and erodes trust in institutions. This, in turn, stifles economic growth by discouraging innovation and job creation. It is crucial to consider these factors and their potential consequences, as highlighted in the literature, in order to ensure a transparent and conducive tax environment that fosters sustainable economic development (Çera et al. 2019; Kiser and Karceski 2017; Kouam and Asongu 2022).

Entrepreneurship, a driving force behind economic growth, sparks innovation, job creation, and overall vitality. Its synergy with taxation forms a pivotal axis within the economic framework. Tax policies shape incentives for entrepreneurs, accentuating their ventures' risks and rewards (Eliakis et al. 2020; Kouam and Asongu 2022). Holcombe's work highlights the link between entrepreneurial activities and economic expansion. Entrepreneurship propels growth through innovation, productivity, and market expansion, with taxation significantly influencing the incentives faced by entrepreneurs (Holcombe 1998). This tax–entrepreneurship nexus is crucial in guiding reforms across labor, capital, corporate, and private wealth taxation. Fundamental to these reforms are the principles of neutrality and modesty within an entrepreneurial tax system (Boozer and Collum 2021; Elert et al. 2019). While our study concentrates on the psychological aspects of tax compliance, recognizing the linkage between taxation and entrepreneurship is essential for a comprehensive view. This underscores the need for well-structured tax policies that incentivize entrepreneurship, foster economic growth, and, consequently, bolster tax revenue (Boozer and Collum 2021; Bruce et al. 2020; Elert et al. 2019; Hedlund 2023; Holcombe 1998).

Our study brings distinctive contributions to the existing literature by deepening the understanding of respectful tax collection behavior's impact on compliance, aligning with previous research while also contextualizing the influence of regulatory changes and legal complexities. Notably, we bridge the gap between economic theory and practice by integrating the tax-neutral principle from the Laffer curve and emphasizing Dan Mitchell's research on the importance of appropriately low taxes for economic growth. Our paper also underscores the consensus against excessive taxation in developing countries. Moreover, we shed light on the symbiotic relationship between entrepreneurship and taxation, emphasizing the principles of neutrality and modesty within entrepreneurial tax systems. Despite acknowledging limitations, our research provides a comprehensive perspective on tax compliance by interweaving psychological dynamics, economic theories, and practical implications, contributing to informed decision-making processes.

While our study offers valuable insights into the correlation between tax officers' behavior and individual taxpayers' compliance in Indonesia, certain limitations must be acknowledged when interpreting and generalizing our findings. The reliance on survey-based data collection, though beneficial in capturing a broad perspective, may introduce biases due to the lack of direct interviewer or mediator control, potentially affecting data depth and accuracy. Furthermore, our research exclusively employed survey forms, omitting in-depth interviews or psychosemantic research methods that could provide richer insights into compliance behaviors. The implementation of these methods, however, requires substantial additional effort and resources. Our study's cultural context, specific to hierarchical norms and deference to authority, should also be considered when extrapolating the findings to other settings. In light of these constraints, we recommend cautious interpretation of our findings. Future research could incorporate diverse methodologies,

particularly integrating in-depth interviews and psychosemantic research, to glean a more comprehensive understanding of tax compliance dynamics. Despite these limitations, our study contributes to the discourse on tax compliance, offering foundational insights and potential implications for practical decision-making processes.

## 6. Conclusions

In this study, we comprehensively examined the influence of tax officers' behavior on individual taxpayers' compliance in Indonesia, revealing the distinct impacts of two behaviors: respectful treatment and authoritarian procedures. We recognize the intricate landscape of tax compliance in a culture marked by hierarchical norms and deference to authority. Our research unequivocally demonstrates that tax officers' adoption of respectful treatment has the profound potential to enhance compliance, foster cooperation, and diminish taxpayer apprehensions, while the use of authoritarian procedures tends to instill intimidation and reduce compliance. Aligned with the Slippery Slope Framework, our study underscores trust-building and power dynamics in shaping compliance behaviors. It is important to acknowledge the limitations inherent in our methodology, relying solely on survey data without direct control over interviewer or mediator biases. In light of these constraints, we approach our conclusions cautiously, advocating for a balanced perspective that highlights the significance of respectful interactions while acknowledging potential complexities. We recommend tax officers prioritize transparent and respectful engagement to foster a compliance-conducive environment. We also emphasize the need for future research to address the identified limitations and enrich the understanding of tax compliance dynamics in the distinct Indonesian context.

**Author Contributions:** Conceptualization, D.P. and E.D.; methodology, D.P.; software, E.D.; validation, D.P. and E.D.; formal analysis, D.P.; investigation, D.P.; resources, E.D.; data curation, D.P.; writing—original draft preparation, E.D.; writing—review and editing, D.P.; visualization, E.D.; supervision, D.P.; project administration, D.P. All authors have read and agreed to the published version of the manuscript.

**Funding:** This research received no external funding.

**Data Availability Statement:** No new data were created or analyzed in this study. Data sharing is not applicable to this article.

**Conflicts of Interest:** The authors declare no conflict of interest.

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
