# Peer review of "Building Trust and Enhancing Tax Compliance: The Role of Authoritarian Procedures and Respectful Treatment in Indonesia"

_jrfm, doi:10.3390/jrfm16080375_

Round 1

Reviewer 1 Report

Please update the literature with new papers, last one is from 2019.

Clearly state the contribution of your paper compared to analyzed literature.

Author Response

Respond to Reviewer

To: Editorial Office

Journal of Risk and Financial Management

Re: Revised Manuscript 

Date: 08 August 2023

————————————————————————————————————

Dear Editorial Office

Thank you for your email dated 06 August 2023 with the reviewer comments for manuscripts entitled “Building Trust and Enhancing Tax Compliance: The Role of Authoritarian Procedures and Respectful Treatment in Indonesia” with manuscript ID jrfm-2543927. As directed by your correspondence, we are submitting the revised manuscripts with changes highlighted in green color that address all reviewer comments and suggestions. Below, please find our responses to each reviewer's comments. We appreciate the opportunity to submit these revised manuscripts and value the feedback provided by the reviewers.

REVIEWER 1:

Reviewer’s Comment

Response

Please update the literature with new papers, last one is from 2019.

Thank you for your valuable feedback regarding the literature in our paper.

In response, we have conducted a comprehensive review of the most recent publications pertaining to the topic. While we have taken steps to incorporate the latest sources, it's worth noting that certain pieces of literature serve as primary sources.

Clearly state the contribution of your paper compared to analyzed literature.

Thank you for your valuable comment regarding the clarity of our paper's contribution compared to the analyzed literature. In response, we have refined our discussion to explicitly articulate how our study advances the existing body of knowledge.

Reviewer 2 Report

I would recommend the conclusions to be modified toward a lesser degree of conviction. The research was elaborated only with one type of instrument, a survey form respondents had to fill in, and as I understand, without control of an interviewer or mediator. One has to take the results from such surveys with more caution.

For more plausible results, deeper methods would be more suitable (in-depth interviews, psycho semantic research etc.) However, this is a totally different type of research and it requires more effort from the researcher.    Nevertheless, the results from the study are a good starting point as a departure for the decision making process, therefore my positive opinion about the publication. 

Author Response

Respond to Reviewer

To: Editorial Office

Journal of Risk and Financial Management

Re: Revised Manuscript 

Date: 08 August 2023

————————————————————————————————————

Dear Editorial Office

Thank you for your email dated 06 August 2023 with the reviewer comments for manuscripts entitled “Building Trust and Enhancing Tax Compliance: The Role of Authoritarian Procedures and Respectful Treatment in Indonesia” with manuscript ID jrfm-2543927. As directed by your correspondence, we are submitting the revised manuscripts with changes highlighted in red color that address all reviewer comments and suggestions. Below, please find our responses to each reviewer's comments. We appreciate the opportunity to submit these revised manuscripts and value the feedback provided by the reviewers.

REVIEWER 2:

Reviewer’s Comment

Response

I would recommend the conclusions to be modified toward a lesser degree of conviction. The research was elaborated only with one type of instrument, a survey form respondents had to fill in, and as I understand, without control of an interviewer or mediator. One has to take the results from such surveys with more caution.

Thank you for your feedback on our manuscript. We have revised our conclusions to reflect this limitation and be more cautious in our statements. We understand the importance of considering these methodological factors in how we present our findings.

For more plausible results, deeper methods would be more suitable (in-depth interviews, psycho semantic research etc.) However, this is a totally different type of research and it requires more effort from the researcher.    Nevertheless, the results from the study are a good starting point as a departure for the decision making process, therefore my positive opinion about the publication.

Thank you for your thorough evaluation of our manuscript and for highlighting the potential for deeper research methods to increase the depth of our findings. We acknowledge the validity of your point regarding the suitability of in-depth interviews and psychosemantic research to produce more comprehensive results. These methods do have the potential to provide richer insights into the subject. However, we have to acknowledge the constraints we face in terms of resources, time and scope. The use of in-depth interviews and psychosemantic research will require a much greater investment of effort and resources. While this approach would have offered a more thorough understanding of the topic, we have opted for a survey-based methodology because of this limitation.

To respond to your comments, we have added these points to the limitations of our study

Reviewer 3 Report

JRFM peer review

August 4, 2023

Dear author(s),

I liked reading your paper, and I think the analysis would benefit Indonesia’s economic development.

Here are some of my suggestions to help you to improve the paper. First, the tax-neutral principle revealed by the Laffer curve is a critical economic theory. I’m surprised the authors didn’t cite and discuss. In this part, I think you need to improve. Because it will help to support your thesis. In addition, Dan Mitchell’s research on promoting economic growth by appropriately low taxes is also critical. I think you need to emphasize that scholars have a consensus that excessive taxation is inappropriate in developing countries. I think you should put this into the public policy recommendations section of the Discussion and Conclusions. Here are the necessary references:

Laffer, A. B. (2004). The Laffer curve: Past, present, and future. Backgrounder1765(1), 1-16.

Edwards, C., & Mitchell, D. J. (2008). Global tax revolution: the rise of tax competition and the battle to defend it. Cato Institute.

Besides, you mentioned the psychological effects of tax policies, such as in lines 44 to 50. This is a good version. However, the psychology-related nudge theory on economic development seems overlooked. It is argued that a good public policy can have a nudge effect on promoting economic growth, which can somehow encourage tax revenue. However, a bad public policy can reverse economic growth and taxes. In this regard, I think in the literature, discussion, and conclusion sections, you need to expand the scope of your analysis a little bit into public policy, economic development, and growth. This can make your paper more attractive. Here are the necessary references:

Espinosa, V. I., Wang, W. H., & Huerta de Soto, J. (2022). Principles of Nudging and Boosting: Steering or Empowering Decision-Making for Behavioral Development Economics. Sustainability14(4), 2145.

Thirdly, lousy tax policy and bad tax collectors’ attitudes can result in corruption, as you mentioned in line 34. Then there will be no economic growth. The above literature also has some insights on that issue. You also need to address the corruption point.

Also, the relationship between entrepreneurship, taxation, and economic growth is overlooked. What is the driving force of the economy? Entrepreneurship and entrepreneurial production. The following references can help you:

Holcombe, R.G. Entrepreneurship and economic growth. Quart J Austrian Econ 1, 45–62 (1998). https://doi.org/10.1007/s12113-998-1008-1

I am looking forward to reading the revised version from you.

Thank you.

Best wishes,

The Reviewer

The English language is acceptable, but it might need minimal proofreading.

Author Response

Respond to Reviewer

To: Editorial Office

Journal of Risk and Financial Management

Re: Revised Manuscript 

Date: 08 August 2023

————————————————————————————————————

Dear Editorial Office

Thank you for your email dated 06 August 2023 with the reviewer comments for manuscripts entitled “Building Trust and Enhancing Tax Compliance: The Role of Authoritarian Procedures and Respectful Treatment in Indonesia” with manuscript ID jrfm-2543927. As directed by your correspondence, we are submitting the revised manuscripts with changes highlighted in blue color that address all reviewer comments and suggestions. Below, please find our responses to each reviewer's comments. We appreciate the opportunity to submit these revised manuscripts and value the feedback provided by the reviewers.

REVIEWER 3:

Reviewer’s Comment

Response

I liked reading your paper, and I think the analysis would benefit Indonesia’s economic development. Here are some of my suggestions to help you to improve the paper:

Thank you very much

First, the tax-neutral principle revealed by the Laffer curve is a critical economic theory. I’m surprised the authors didn’t cite and discuss. In this part, I think you need to improve. Because it will help to support your thesis. In addition, Dan Mitchell’s research on promoting economic growth by appropriately low taxes is also critical. I think you need to emphasize that scholars have a consensus that excessive taxation is inappropriate in developing countries. I think you should put this into the public policy recommendations section of the Discussion and Conclusions. Here are the necessary references:

Laffer, A. B. (2004). The Laffer curve: Past, present, and future. Backgrounder, 1765(1), 1-16.

Edwards, C., & Mitchell, D. J. (2008). Global tax revolution: the rise of tax competition and the battle to defend it. Cato Institute.

I appreciate your insightful comment regarding the significance of the tax-neutral principle and Dan Mitchell's research in shaping our study. We have certainly incorporate these aspects into the public policy recommendations section of our Discussion and Conclusions, as it resonates with the broader implications of our study.

Besides, you mentioned the psychological effects of tax policies, such as in lines 44 to 50. This is a good version. However, the psychology-related nudge theory on economic development seems overlooked. It is argued that a good public policy can have a nudge effect on promoting economic growth, which can somehow encourage tax revenue. However, a bad public policy can reverse economic growth and taxes. In this regard, I think in the literature, discussion, and conclusion sections, you need to expand the scope of your analysis a little bit into public policy, economic development, and growth. This can make your paper more attractive. Here are the necessary references:

Espinosa, V. I., Wang, W. H., & Huerta de Soto, J. (2022). Principles of Nudging and Boosting: Steering or Empowering Decision-Making for Behavioral Development Economics. Sustainability, 14(4), 2145.

We genuinely appreciate your insightful feedback on our manuscript and your valuable recommendations to enhance its comprehensiveness and appeal.

To address your suggestion, we have expand the scope of our analysis to incorporate discussions on the nudge theory, public policy's role in economic development, and growth. This expansion will enrich the manuscript by providing a more comprehensive perspective on the relationship between tax policies, decision-making, and economic outcomes.

Thirdly, lousy tax policy and bad tax collectors’ attitudes can result in corruption, as you mentioned in line 34. Then there will be no economic growth. The above literature also has some insights on that issue. You also need to address the corruption point.

Thank you for your insightful feedback.

We appreciate your input, which has led us to realize the importance of addressing this aspect more explicitly. In our revised version, we have definitely expand our discussion to include this relationship.

Also, the relationship between entrepreneurship, taxation, and economic growth is overlooked. What is the driving force of the economy? Entrepreneurship and entrepreneurial production. The following references can help you:

Holcombe, R.G. Entrepreneurship and economic growth. Quart J Austrian Econ 1, 45–62 (1998). https://doi.org/10.1007/s12113-998-1008-1

Thank you for your insightful comment on our manuscript. We appreciate your recommendation and the reference you provided, This has highlighted a valuable aspect that we unintentionally overlooked. And we have incorporated the insights from this topic and reference into our discussion as you suggested.

The English language is acceptable, but it might need minimal proofreading.

Thank you for your feedback on the acceptability of the English language used in our paper. We have conducted a thorough proofreading to ensure the highest level of clarity and correctness in our writing.

Round 2

Reviewer 3 Report

Dear authors,

For my part, the paper is ok to be published. I think the structure is excellent now based on the peer review opinions.

Congratulations!

Best,

The reviewer

No judgment.